# Online Museums Segmentation with Structured Data: The Case of the Canary Island's Online Marketplace

**Gonzalo Díaz Meneses** [1,*], **Miriam Estupiñán Ojeda** [1] **and Neringa Vilkaité-Vaitoné** [2]

1 Department of Economics and Business, University of Las Palmas de Gran Canaria, The Canary I, 35017 Las Palmas de Gran Canaria, Spain; miriam.estupinan101@alu.ulpgc.es
2 Department of Management, Vilnius Gediminas Technical University, 10223 Vilnius, Lithuania; neringa.vilkaite-vaitone@vilniustech.lt
* Correspondence: gonzalo.diazmeneses@ulpgc.es

**Abstract:** This paper's primary objective is to segment the online marketplace of the Canary Islands' museums by using different conversion funnel metrics. Little systematic research exists on digital user behaviour, and much less is known about how to segment cultural users with structured data from manually extracted and SEO software sources. With this aim in mind, we built a database with data related to the different phases of the conversion funnel of the museums to segment this online museum marketplace. In the findings, not only do we acknowledge the existence of different segments, but we also provide insight into the user's digital behaviour by considering different metrics from the different phases of the conversion model process (awareness, consideration, conversion and loyalty). The originality of this paper is multifold. Firstly, it estimates the potential optimisation of these websites to improve the digital marketing implemented by the museum sector of the Canary Islands. Secondly, it sheds light on what benchmarking tactics and statistics procedures can be followed to carry out a non-hierarchical segmentation with standardised and comparable data. Thirdly, it contributes to the literature of digital marketing by eclectically combining the conversion funnel model, benchmarking techniques and non-hierarchical segmentation procedures.

**Keywords:** segmentation; museum; online user behaviour; digital marketing

## 1. Introduction

In the face of the COVID-19 pandemic, many governments have implemented strict measures to limit the spread of the virus. In many countries of the world, cultural organisations, including museums, were forced to stop their activity at least temporarily. The Canary Islands were no exception. Museums that for the last few decades were intensively adopting information and communication technology channels, innovations that seek to fulfil the needs and expectations of customers [1,2], have, since March 2020, had restrictions on the possibilities to motivate customers to move along the conversion funnel to purchase, depending on the alert level. Currently, when there exists a prospect for an uninterrupted activity of cultural entities due to the increasing vaccination, museums strive for every visitor and implement measures in order to overcome challenges brought by decreased attendances, limited resources and increased competition [3]. Despite the optimistic statistics in the field of vaccination, new variants of the COVID-19 approach (at the time of this research, Delta) dominate the scene. In these difficult times, when there exists a requirement to ensure social distancing at large scales to prevent the spread of the virus, websites as digital sales channels gain a priority in marketing communication of museums [4].

The pandemic has made us give credit to digital marketing techniques and urged us to optimise the online marketplace everywhere. Perhaps before the Covid crisis, we should have put more emphasis on improving the Canary Islands' museums websites and how the destination managed user traffic, consideration, online purchases and loyalty for the

museums' ecosystems, but it was not until now that we realised just how crucial a task it is. How much traffic is the Canary Islands' cultural online spaces attracting? What is the consideration phase potential of the online museums? Are these museums oriented towards the purchase funnel phase so that they are boosting online sales? Are the museums' websites enhancing user loyalty? Can we segment the online marketplace of the Canary Islands' museums? Can we segment the Canary Islands' museums by using the conversion funnel metrics as criteria? These are the questions that need to be raised, and this paper is dedicated to answering them.

Usually, market segmentation based on status in the conversion funnel covers only customers [5–7], while the organisations (important members of the market as well) are left in touch using the other models and criteria for segmentation, for example, the category of the service provider, persuasiveness [8] and pricing [9]. Recent studies have examined the problems of website segmentation in hotel chains [8], dating platforms [9] and cruise operators [10]. Little systematic research exists on digital user behaviour according to the conversion funnel stages [11–13], and much less is known about how to segment cultural users with structured data from manually extracted and SEO software sources. While there are sporadic research works of websites of museums [1,14–16], empirical validation of online museums' segmentation thereof is still scarce. To the best of our knowledge, the same holds for research examining the conversion funnel for segmenting the online marketplace of museums.

This paper's primary objective consists of segmenting the online marketplace of the Canary Islands' museums by using different conversion funnel metrics. These conversion funnel metrics are employed as segmentation criteria to measure the optimisation potential of the attraction, consideration, purchase and loyalty conversion phases of these cultural spaces. Furthermore, it aims at distinguishing different museum segments to improve the efficacy and efficiency of the digital marketing of policymakers. This type of research has not been conducted before.

The theoretical framework of this paper combines the conversion funnel model with particular metrics for each phase of the funnel. By analysing Canary Islands' museums through the prism of a conversion funnel, this paper contributes to the understanding of segmentation with structured data in the context of museums as cultural organisations that are a considerable part of the underrepresented sector in electronic commerce literature. This type of analysis, which allows one to identify status in the conversion funnel, has not been conducted before. Hopefully, the findings will help managers and curators of the Canary Islands' museums to appreciate the impact of a well-functioning website that leads to customer loyalty and to make decisions regarding resource allocation and the development of online marketing strategies. The findings have the potential to be used to fine-tune the measured websites to provide customers with more engaging content and to improve the customer path that leads to the loyalty phase.

The paper is divided into four sections: the theoretical section relates to the review of the literature, touching upon the definitions of the conversion funnel stages and pinning down the most suitable metrics for the upcoming segmentation; the methodology section includes the description of the measuring instruments, survey and other procedures; and the analysis of the results section encompasses the empirical evidence derived from some descriptive statistical data and the K means output. Finally, we discuss theoretical and practical implications with limitations and future lines of research and draw some conclusions.

## 2. The Review of the Literature

### 2.1. The Conversion Funnel and Its Phases

The conversion funnel, also referred to as a marketing funnel, has been at the epicentre of the marketing literature and research for the past years. The funnel model is widely used in the area of marketing and sales practitioners as it is very useful to postulate how consumers behave throughout the decision-making process [17–19]. During this process,

customer selects a museum to visit, satisfying a purchase intention and purpose of visiting, minimizing a perceived risk and maximizing the benefits of visiting [7].

The conversion funnel model as we know it today can be defined as the strategy through which a company determines the phases to get in touch with potential customers in order to achieve a final purpose, which can be closing a sale or getting a loyal customer [20]. In the digital marketing field, the author of [21] states that the conversion funnel refers to a methodology that defines the journey that users take from when they visit a website to when they reach the final conversion goal.

Marketing has mostly associated the above-mentioned process with information processing theory [18,22]. This theory is the basis of most customer behaviour models and suggests that customers must go through different reflection stages during the purchase decision process until reaching the final decision: (1) problem or need recognition, (2) search for product information, (3) evaluation time, (4) purchase and (5) post-purchase satisfaction [17,22,23]. This also makes it similar to the widely applied AIDA model (from its acronym attention, interest, desire and action), developed in 1898 by St. Elmo Lewis [24]. The aforementioned model was designed as a framework for planning and evaluation of persuasive marketing communication efforts [25].

The utility of the conversion funnel is vast. First of all, it is a fundamental process to understand how the customer behaves, as well as being a crucial method for companies and advertisers to know how to sell a product or a service [26]. Secondly, the structuring device of the model makes it very useful [19]. Its funnel-shaped representation makes it very easy to understand and visualise the customer's entire journey. In this vein, the funnel is wide at the top and narrow as its bottom is reached. The sales funnel narrows as customers go through it, representing the potential loss of customers at the lower parts of the funnel [27,28].

Although there are several variants of the conversion model process that have been proposed over the last few years, the most widely accepted and used model has four basic phases, or stages: awareness, consideration, conversion and loyalty [22,29–32]. This labelling scheme is precisely the one we used in this research.

According to this model, the conversion funnel starts with the 'awareness'—or attraction—phase. This first stage is maybe one of the most important, not only because it is at the top level of the funnel and is the starting point of the whole process but more importantly because it is the stage in which the customer is aware of the existence of a product, service or brand [30] that can solve a certain need or desire [22,33]. If the customer is exposed, for example, to an advertisement or lands on a website, this state of awareness can be generated [17]. Therefore, the objective of every company is basically to generate sufficient brand awareness [1,34,35] through the creation of interesting and attractive web content. We can even define this brand awareness as simply the ability of the potential customer to remember a brand and associate it with a specific category or product line [36]. Then, it is necessary to gain presence, introduce the brand to potential customers and capture their attention [27]. In the case of museums, this is of considerable importance because 70% of museum visitors specifically look for information about museums online prior to visiting [14].

Once the customer is aware of the existence of a brand, they will try to acquire more knowledge about a specific product or service [22]. The customer will consider various options to buy from [33]. Thus, the customer conversion funnel continues with 'consideration'—often called the research phase. During this stage, it is normal for the customer to view a specific product page in more detail. This is where the customer's interest can be peaked, potentially leading to a willingness to carry out the purchase action [30]. The main goal of the marketer during this second stage is to give users as much information as possible and persuade them to remain on the website long enough for them to seriously consider the brand against its competitors [37]. Good performance at the consideration stage is based on providing useful information, generating good leads and interacting with content on social media [29].

The third stage is 'conversion'—often called the purchase or action phase. Only those customers who are truly interested in the purchase will remain in this stage of the funnel [32]. The conversion stage has often been defined as a key phase in which the customer takes action and completes the transaction. This stage encompasses the whole purchase process, from when a decision is clear until the purchase order is finally made [33]. It is important to provide the necessary facilities and information to customers so that they can carry out the purchase action by, for example, making the payment and checkout process as simple and as quick as possible in order to provide a satisfying experience [38]. The choice of a strategy that has a focus upon either quality or promotion helps stimulate the conversion as well [39].

Finally, the last step in the funnel is related to 'loyalty' and post-purchase satisfaction indicators. This last stage is sometimes referred to as the *retention* phase, where the customer becomes an evaluator and has the opportunity to advocate on behalf of the brand [29,40]. According to [41], it is imperative to engage the customer to inspire and cultivate loyalty. So, the main goal in this final stage is to establish loyalty programmes, customer communities or social platforms to create strong and close relationships [29,32]. In the case of museums, customer loyalty can be strengthened by a well-designed website [1] and the participatory voices of museum curators [16].

Although those four steps may seem like an easy process, a multitude of factors influences the customer's journey from becoming aware of the product to making the final decision to purchase [42]. A significant number of people will know that a brand or a product exists; only some of them will be truly interested—even fewer will become customers by making a purchase. Finally, only a small fraction of those will be completely satisfied and develop a high degree of loyalty [19]. The reality is that only a few visitors complete the transaction, and only those customers who have fully consolidated the previous stage go to the next level [37]. In an ideal model, the customer moves from the top of the funnel to the middle phases and finally at the bottom. In the digital journey, the lines between the four phases have become blurred. It means that the customer can enter the conversion funnel at any of four phases and go back and forth across the phases [33]. Proper management and understanding of the funnel make it easier to spot potential gaps where customers drop out and never convert; entrance phases; and moving directions.

### 2.2. The Conversion Funnel Metrics as Criteria to Segment

The conversion funnel model enables tracking customer behaviour throughout the sales process with the help of properly chosen metrics [5]. In academic and professional literature [43–45], we can find many examples of metrics and techniques to measure the different stages of the conversion process. Selection of the proper combination of metrics as criteria to segment the online marketplace is a challenging task even for the most sophisticated marketers [46]. After reviewing several research works, we selected the most suitable metrics for the segmentation task.

The first stage of the conversion funnel, i.e., attraction, will be measured with one of the most efficient consumption metrics to measure brand awareness: traffic generation. Through this metric, it is also possible to know the number of visits to a website and its significance [45,47]. Measurement of visitor statistics, according to [44], is a core activity for any business. Although there are several techniques for collecting a website's traffic data, we will use the one in which data is provided directly by internet service providers—webmasters [47]. Using this technique is a great advantage since the traffic of a large number of websites can be collected on the same site [48]. On this basis, the first hypothesis is put forward as follows:

**Hypothesis 1.** *The attraction metric of traffic might be used for describing the segments of the museums' online users.*

In the second stage, that is, the consideration conversion phase, links to social networks are of considerable importance because, at the consideration phase, social media has

a huge potential to increase traffic, engagement, encourage innovation, and ideation, and to generate leads [29,49]. Efficient marketing communication in social networks shifts the customer to the purchase stage of the conversion funnel but by making the consideration stage longer inasmuch as it causes a zero moment of truth [50]. Consequently, the active links to social networks are of considerable importance since they provide information [51] and share people experiences [52]. Social media engages consumers and makes the involvement stage of the decision journey more important in the context of music festivals [53] and museums [54]. Likewise, on each web page, we measure the existence of links to official profiles on social networks [55] because social media can impact positively or negatively segment different groups during the consideration phase [56]. This influence of social media is through conversations and word of mouth [56], and it addresses a two-way contact in online museum platforms [54]. On this basis, Hypothesis 2 is proposed as follows:

**Hypothesis 2.** *The interest metric of social media links on websites might be used for describing the segments of the museums' online users.*

Blogs afford cultural organisations the opportunity for direct and personal interaction with audiences [55]. The high credibility of blogs as informational sources [52,57] might strengthen the intensity of the customer journey through the purchase funnel. In addition to being an informational source, blogs are crucial for intrinsic and affective segments [58]. Therefore, blog entries are treated as the interest metric that might be used for describing online users [53], and Hypothesis 3 is put forward as follows:

**Hypothesis 3.** *The interest metric of blog entries per website might be used for describing the segments of the museums' online users.*

The download function is the URL address to the download path [59]. The possibility to download interactive resources might lead to website satisfaction [60], so the interest of the user can be awakened as well. There is the possibility of downloading interactive resources such as maps or museum guides since it has been shown that the user is more likely to stay in the conversion funnel and consolidate the consideration phase if the website offers a large number of participation options [55]. Moreover, the use of tourism apps is even becoming relevant for certain senior segments [61], and where mobiles are concerned, it can mediate the traveller experience [62]. Likewise, museum visitors are likely to download online images of artifacts and a wide range of resources such as educational and research materials [63]. Every so often, these resources are suitable for mobiles [64] and include location and weather functionalities [65]. That said, we can put forward Hypothesis 4 as follows:

**Hypothesis 4.** *The interest metric of download per website might be used for describing the segments of the museums' online users.*

Finally, we can actively measure how museums' websites manage the consideration phase through the analysis of some visit leads. The use of these informational features can facilitate the consideration of the brand and indicate the level of digital transformation maturity as well as the acceptance of technology by the different segments [66]. Some research even indicates that online behavioural intentions are highly influenced by levels of satisfaction with the information available on a website [67]. This is the pre-purchase phase, so more information options about the characteristics of the site can significantly influence a later decision of choice. Likewise, visitors rely on the museum website to answer questions after completing forms [63]. Usually, websites consistently provide information about business hours, addresses and contact information [68]. In the conducted research of cultural organisations, the contact telephone number, contact email address, forms to request more information and information about opening hours are some of the selected

metrics to measure visit leads [69] and cultural participation [54]. In line with this, we can put forward Hypothesis 5 as follows:

**Hypothesis 5.** *The interest metric of visits leads per website might be used for describing the segments of the museums' online users.*

The third level, according to the conversion stage, is measured based on the existence of some purchase leads [54]. The main goal is to measure the sales potential of the Canary Islands' museum websites and how effectively they convert users. Some of the important items for evaluating websites in this phase include information regarding the existence of a souvenir shop, the possibility of buying some souvenirs online and the opportunity to book or buy tickets online [63]. What is more, the lack of online ticketing policies allows one to distinguish a group of dissatisfied museum visitors. No doubt, booking and ticketing online reduce paper work and imply the existence of a smart museum with segmentation capabilities by considering different types of requirements for children, adults, seniors, etc. [70–72]. We talk about some signals and buttons that facilitate the conversion phase as long as they are useful metrics to identify the efforts of websites to achieve customer conversion. Taking it into account, the sixth hypothesis is put forward as follows:

**Hypothesis 6.** *The purchase metric of purchasing and booking links per website might be used for describing the segments of the museums' online users.*

In the last part of the funnel, the priority is to find out how the Canary Islands' museums manage their digital presence to create close relationships. In accordance with [73], lead generation metrics are some of the most effective metrics for measuring customer–company relationships and post-purchase loyalty. With this in mind, lead generation allows us to know how the customer perceives certain content and how they interact with it, being a clear demonstrator of their level of satisfaction and relationship with the brand. Thus, efforts are focused on analysing and identifying the existence of some data collection mechanisms such as using interactive story telling techniques [74,75] and generating leads such as subscription newsletters [54], form completions [76,77] or the possibility of getting opinions and new ideas from users through the existence of the complaint and claim forms [78] as well as by managing community platforms [79]. On this basis, Hypothesis 7 is put forward as follows:

**Hypothesis 7.** *The metric of loyalty leads per website might be used for describing the segments of the museums' online users.*

As a result of the above, the use of the cited metrics in each of the four phases of the funnel allows us to research how users of the ecosystem of digital museums in the Canary Islands behave and how every segment can be profiled.

### 2.3. The Interplay between Online and Offline Spaces

Before starting to evaluate the digital management of Canary Island's museum websites, a phenomenon that can cause an alteration in the results must be taken into account. We mention precisely the interplay between online and offline spaces in which users are under the influence of their physical context. On many occasions, context attributes of the museum, especially physical location, have proven to be important elements because they act as factors that strongly influence the motivation to visit a particular place [80]. In other words, the intentions to visit an establishment are conditioned by its physical location [80], and there are even many visitors who seek to feel a big connection with the cultural location before visiting a museum [81]. In this line, the quality of a location, understood as a central physical placement that is easy to find and that is not so far from other centres of interest, positively influences visitor preferences [82]. Accordingly, it is highly probable that those museums that are located in more universal, accessible and

relevant destinations are going to receive more physical visits as they are more likely to be known and, therefore, they must present different (and higher) digital metrics due to the greater attraction, interest, loyalty and commitment that their physical and cultural location arouses. By contrast, those other cultural centres located in unknown, regional or limited territories are likely to have fewer offline visits and, therefore, poorer visits and lower interest and loyalty rates on their online spaces. This is so because, before physically visiting a museum, users frequently visit its website to obtain detailed information [83], and, therefore, the volume of real and physical visits largely determines the magnitude of visits to the website, even if this online visit is prior to the physical visit.

This phenomenon might occur in the territory of the Canary Islands, made up of seven islands of different relevance, impact and size. For example, brick and mortar museum buildings located on more remote or unknown islands (El Hierro, La Gomera, Lanzarote or La Palma) would receive fewer physical visits and, therefore, would present lower digital metrics. However, the museums located on islands that are better known, more inhabited or with greater connections with foreign countries (Tenerife, Gran Canaria or Fuerteventura) would obtain more visits to their website as a consequence of a greater number of physical and real visits.

Given that, part of the interest and commitment that users show with Canary Island's museum websites could be the result or consequence of the influence of the physical museum dimension, understood as the tourist destination and its characteristics of popularity, remoteness or relevance. This continuous interaction between physical context and digital environment, which is not independent, suggests that there could be a holistic relationship between offline and online spaces [84] and causes many museums to develop their digital presence as an expansion of their physical establishment [85].

In accordance with all the above, the characteristics of the tourist destination and especially on which island the museum is physically located are revealed as critical elements for determining the digital behaviour of visitors on their respective websites. On this basis, Hypothesis 8 can be put forward as follows:

**Hypothesis 8.** *The island on which the museum is located relates to the museum segment metrics.*

### 3. Materials and Methods

A database is designed to gather information related to the Canary Islands' Museums websites. This database shows 11 variables and 155 cases. Concerning the variables, the first three variables refer to the museum name, URL and island. Moreover, seven variables are dedicated to metrics about the different phases of the conversion funnel—that is, attraction, consideration, conversion and loyalty. For the attraction phase, we consider traffic and the number of users visiting the websites monthly. For the consideration phase, we contemplate the existence of blog, social media and download resources. Within the consideration phase, we include visit leads to collect data about the presence of phone and email contacts, opening times, bricks and mortar addresses, and any form to be filled with further information. For the conversion funnel, we encompass a variable labelled purchase dimension, whose values increase if there is information about a souvenir shop, and the possibility of buying souvenirs or booking or buying entrance tickets for the museum itself or any event that might take place there. For the loyalty phase, the variable scores the existence of newsletters, forums and club subscription links. Likewise, if there is a suggestions or complaints mailbox, as long as there is any channel to receive a phone call or further information, this variable's value rises. Finally, there is a variable to identify the survey taker and database builder.

Thanks to possessing an exhaustive list of the Canary Islands Museums [86], ninety researchers carried out a simultaneous web analytic task for these 'cultural space' websites in May 2021. This survey was performed in four sessions under the supervision of the head researcher, although the team of researchers were located in different places with a connection through TEAMS. So, a database was built with information related to different

variables that fall into the different phases of the conversion funnel. Firstly, we made good use of Ubber Suggest software to estimate the traffic of these websites. Secondly, we manually examined the websites to complete the consideration, conversion and loyalty funnel model metrics. As a result, the final database comprised just 68 cases due to the fact that not all the museums have websites or their website was not working during the survey, whose period was 11–14 May 2021.

Tables 1 and 2 layout the sampling frame list classified by island.

**Table 1.** El Hierro, Fuerteventura, La Gomera, Lanzarote and La Palma's museums.

| EL HIERRO | |
|---|---|
| Geological Interpretation Centre | Interpretation Centre of El Julan Cultural Park |
| Volcanological Centre | Guinea Ecomuseum |
| Ethnographic Center Casa de las Quinteras | Centre of the Biosphere Reserve |
| El Garoe Interpretation Centre | |
| **FUERTEVENTURA** | |
| Interpretation Centre Village of Atalayita | Archaeological Museum of Betancuria |
| Interpretation Centre of Cueva del Llano | Interpretation Centre of Jandia Natural Park |
| Juan Ismael Art Centre | Salt Museum of Salinas del Carmen |
| Casa Mané Art Centre | La Cilla Grain Museum |
| Doctor Mena House-Museum | La Alcogida Ecomuseum |
| Unamuno House-Museum | House of the Colonels |
| Morro Velosa Viewpoint and Interpretation Centre | Interpretation Centre of Tiscamanita |
| Lighthouse of La Entallada | Tradicional Fishing Museum |
| **LA GOMERA** | |
| Ethnographic Museum of La Gomera | Ethnographic Park of La Gomera—Los Telares |
| Archaeological Museum of La Gomera | |
| **LANZAROTE** | |
| International Museum of Contemporary Art | Aeronautical Museum of Lanzarote Airport |
| Cesar Manrique Foundation | Chinijo Museum |
| Visitors and Interpretation Centre of Mancha Blanca | César Manrique Haría House-Museum |
| Tanit Ethnographic Museum | The Timple House-Museum |
| Museum of Piracy | Yaiza Aloe Vera Museum |
| Punta Mujeres Aloe Vera Museum | El Grifo Museum of Wine |
| El Patio Agricultural Museum | Jose Saramago House-Museum |
| Atlantic Museum | Museum-Information Point Echadero de los Camellos |
| **LA PALMA** | |
| Interpretation Centre of the Marine Reserve | Ethnographic Museum of José Luis Lorenzo Barreto |
| Museum-Dressing Room of the Virgin of Las Nieves | Sacred Art Museum |
| Belmaco Archaeological Park | Casa del Maestro Ethnographic Museum |
| Interpretation Centre of La Bajada | Benahoarita Archaeological Museum |
| The Silk Museum | The Red House |
| Banana Museum | The Waffle Interpretation Museum |
| Naval Museum—Boat of the Virgin | Insular Museum of La Palma |
| La Zarza and La Zarcita Cultural Park | Puro Palmero Museum |
| El Tendal Archaeological Park | Interpretation Centre of Sugar Cane and Rum |
| Casa Luján Ethnographic Museum | The Plains of Aridane |
| City in the Museum of Contemporary Art Forum | Las Manchas Wine Museum |
| Visitors and Interpretation Centre of Cumbre Vieja Natural Park | Visitors Centre of Caldera de Taburiente National Park |

**Table 2.** Gran Canaria and Tenerife's museums.

| GRAN CANARIA | |
|---|---|
| La Rama Museum | Municipal Museum of Arucas |
| The Martin Chirino Art and Thought Foundation | Casa los Yanez Ethnographic Museum |
| Tejeda Medicinal Plants Centre | Maipes de Agaete Archaeological Park |
| Antonio Padron House- Museum—Indigenous Art Centre | Canarian Museum |
| Finca Condal Museum | Abraham Cardenes Sculpture Museum |
| Triptych of Our Lady of the Snows | Cueva Pintada Archaeological Park and Museum |
| Mata Castle Museum | Panchito Ecomuseum |
| Museum of History and Traditions of Tejeda | Aguimes Interpretation Centre |
| Interpretation Centre of Casa del Fraile | Diocesan Museum of Sacred Art |
| La Zafra Museum | Leon and Castillo House-Museum |
| Interpretation Centre of Guayadeque | Ingenio Interpretation Centre |
| Elder Museum of Science and Technology | La Fortaleza Interpretation Centre |
| Aeronautical Museum of Canary Islands Air Command | Canary Museum of Meteorites |
| The Canary Island Craft and Stone Museum | Canary Islands Naval Museum |
| Interpretation Centre of El Pastoreo | Gando Tower Museum |
| Aguimes History Museum | Living Museum of La Aldea |
| Nestor Museum | Interpretation Centre of Las Salinas de Tenefe |
| Patrons of the Virgin House-Museum | Lugarejos Locero Centre |
| Colon House | Domingo Rivero Museum |
| Gofio Museum | Camarín Virgen del Pino Museum |
| Ethnographic Museum of Casas Cuevas | Perez-Galdos House-Museum |
| Lomo de los Gatos | Necropolis of Arteara |
| Ethnographic Centre of Valleseco | Labrante Interpretation Centre |
| Atlantic Centre of Modern Art | Tomas Morales House-Museum |
| Cenobio de Valerón | Arehucas Rum Distillery |
| La Regenta Art Centre | Interpretation Centre of El Paisaje |
| Nestor Alamo Museum | |
| TENERIFE | |
| Tenerife Aloe Vera Museum | Manuel Martín González Museum Room |
| Pinolere Ethnographic Museum | Science and Cosmos Museum |
| The Canary Islands Military History Museum | La Baranda Wine House |
| El Quijote Museum | Archaeological Museum of Puerto de la Cruz |
| Museum of Education of the University of La Laguna | Municipal Museum of Fine Arts |
| Sierva de Dios House-Museum | Ethnographic Guimar Pyramid Park |
| Eduardo Westerdahl Museum of Contemporary Art | Canarian Institute Museum—Cabrera Pinto |
| Nature and Man Museum | Sacred Art Museum of San Marcos' Church |
| Sacred Art Museum 1 | Sacred Art Museum 2 |
| Natural Museum of Educational Mathematics | The Tenerife History and Anthropology Museum |
| Rooms Arts of the Government of the Canary Islands | The Carpet Museum |
| Los Sabandeños House-Museum | Captain's House Museum |
| Tenerife Arts Space | History Museum Granadilla de Abona |
| Museo Sacro El Tesoro de La Concepción | Cristino de Vera Foundation |
| Castillo de San Cristóbal Interpretation Centre | Cha Domitila Ethnographic Museum |
| Juan Évora Ethnographic Centre | Ibero-American Craft Museum of Tenerife |
| Interpretation Center of the I and II International Exhibition of Sculptures on Street 1973 and 1994 | Santa Clara Museum of Sacred Art |
| Fisherman's Museum | Zamorano's House |

As can be seen in Table 3, the sampling procedure was probabilistic at random and the error reaches 2%.

**Table 3.** The survey spreadsheet on The Canary Islands museums.

| | |
|---|---|
| PROCEDURE | Probabilistic at random <2% error with 95% reliability |
| POPULATION | Museums and cultural spaces in The Canary Islands |
| ELEMENTS | Websites of 155 museums |
| SAMPLING FRAMEWORK | List of museums provided by Dirección General de Patrimonio del Gobierno de Canarias |
| SAMPLING UNITS | The Canary Islands Museums websites |
| SAMPLING SIZE | 68 websites |
| CONTACT | Desktop, laptops and tablets with the internet connection used by 90 students of ULPGC |
| DATE | 4 and 6 May 2021 |
| PLACE | From the students' households |
| CONTROL SYSTEM | Synchronic connection by TEAMS and submission through the ULPGC Virtual Campus |

## 4. Results

### 4.1. Preliminaries

Prior to distinguishing different segments, we carried out descriptive statistics in order to represent the conversion funnel of the Canary Island museum marketplace. To be specific, it can be said that the number of users of the sampling units reached 252,540. As this represents more than fifty per cent of the total museums' population, it can be assumed that the total traffic of the online museums' marketplace must be over 300,000 users per month (see Figure 1).

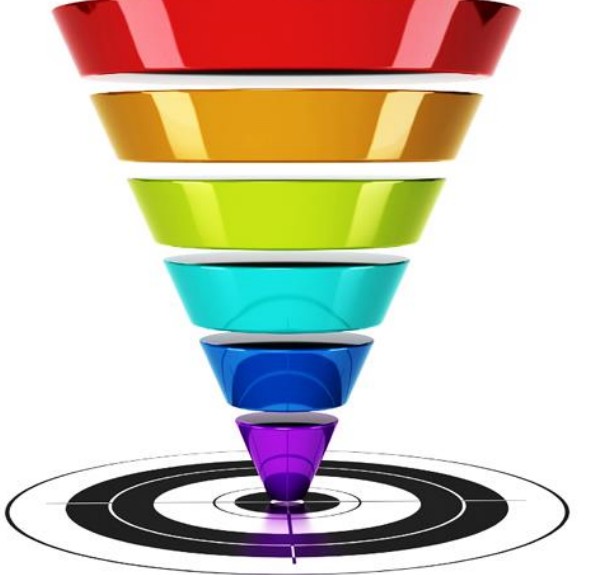

**ATTRACTION**

Average traffic on websites monthly: 252,540 visits

**INTEREST**

Blog on average: 0.19

Social media links on websites: 2

Download on average per website: 1

Visit forms per website on average: 3.15

**PURCHASE**

Purchasing and booking links on average per website: 0.88

**LOYALTY**

Loyalty leads on average per website: 0.59

**Figure 1.** The Canary Islands museums conversion funnel.

Regarding interest, it seems clear that while the museums only show two social media entries and one link to download a file, they display more than three visit forms on average.

Similarly, we can point out that the museums show less than one purchasing and booking link per website. Finally, only half of the museums make some effort to retain the loyalty of their visitors. A nuanced disclosure of the descriptive statistics as regard the Canary Islands' museums websites is laid out in Table 4.

**Table 4.** Frequency and descriptive statistics of the Canary Islands' museums websites.

| Descriptive Statistics | | | | | |
|---|---|---|---|---|---|
| | N | Minimum | Maximum | Mean | Std. Deviation |
| Traffic | 68 | 0 | 14,380,075 | 252,540.26 | 1,742,765.305 |
| Blog entries | 68 | 0 | 1 | 0.19 | 0.396 |
| | | 0 blog = 55 (80.9%); 1 blog = 13 (19.1%) | | | |
| Social Media | 68 | 0 | 7 | 2.03 | 1.931 |
| | 0 social media = 20 (29.4%); 1 social media = 13 (19.1%); 2 social media = 12 (17.6%); 3 social media = 5 (7.4%); 4 social media = 8 (11.8%); 5 social media = 7 (10.3%); 6 social media = 2 (2.9%); and 7 social media = 1 (1.5%) | | | | |
| Downloads | 67 | 0 | 16 | 1.03 | 2.516 |
| | 0 download = 44 (64.7%); 1 download = 11 (16.2%); 2 downloads = 5 (7.4%); 3 downloads = 1 (1.5%); 4 downloads = 2 (2.9%); 5 downloads = 1 (1.5%); 6 downloads = 1 (1.5%); 10 downloads = 1 (1.5%); and 16 downloads = 1 (1.5%) | | | | |
| Visit leads | 68 | 0 | 6 | 3.15 | 1.558 |
| | 0 visit leads = 7 (10.3%); 1 visit lead = 7 (7.4%); 2 visit leads = 8 (11.8%); 3 visit leads = 10 (14.7%); 4 visit leads = 28 (41.2%); 5 visit leads = 9 (13.2%); and 6 visit leads = 1 (1.5%) | | | | |
| Purchase leads | 68 | 0 | 10 | 0.88 | 1.792 |
| | 0 purchase leads = 44 (64.7%); 1 purchase lead = 10 (14.7%); 2 purchase leads = 8 (11.8%); 3 purchase leads = 2 (2.9%); 6 purchase leads = 3 (4.4%); and 10 purchase leads = 1 (1.5%) | | | | |
| Loyalty Leads | 68 | 0 | 4 | 0.59 | 1.026 |
| | 0 loyalty leads = 46 (67.6%); 1 loyalty lead = 11 (16.2%); 2 loyalty leads = 6 (8.8%); 3 loyalty leads = 3 (4.4%); and 4 loyalty leads = 2 (2.9%) | | | | |

Finally, we standardised the values of these variables before using them to segment. In this way, comparisons and interpretation are much easier.

### 4.2. K Means Results

Intending to distinguish different segments of museums, we carried out a k means analysis (see Table 5). Three different segments were pinned down after discarding two and four groups seeing that the Anova test showed non-significant metrics and after thinking that the triple segment structure is the most suitable for managerial implications. What is more, this triple segment structure seemed reliable given that 100% per cent of cases were correctly classified in light of the discriminant test. Moreover, this segment solution was stable because convergence was achieved and there were no changes in the cluster centroids after six iterations. Furthermore, it is worth noting that the minimal distance obtained between initial centroids was 8.9. As can be seen in Table 5, the first segment gathers 46 museums whose online performance shows room for improvement. The second segment gathers 20 museums whose number of social media, download and purchase leads is outstanding, although their traffic is improvable. Finally, there is a segment with only one museum—that is, the Saramago museum, which sets an example in terms of traffic, blog entries, visit leads and loyalty leads.

**Table 5.** K means analysis and segmentation results of the Canary Islands' museums websites.

| | **Final Cluster Centers** | | |
| --- | --- | --- | --- |
| | **Cluster** | | |
| | **1** | **2** | **3** |
| Traffic | −0.11701 | −0.12907 | 8.10639 |
| Blog entries | −0.26308 | 0.52713 | 2.04170 |
| Social media | −0.47670 | 1.02027 | 0.50252 |
| Downloads | −0.26239 | 0.60409 | −0.01186 |
| Visit leads | −0.22004 | 0.41922 | 0.54763 |
| Purchase leads | −0.32262 | 0.79127 | −0.49249 |
| Loyalty leads | −0.46758 | 0.93778 | 2.35163 |

| | **ANOVA** | | | | | |
| --- | --- | --- | --- | --- | --- | --- |
| | **Cluster** | | **Error** | | **F** | **Sig.** |
| | **Mean Square** | **df** | **Mean Square** | **df** | | |
| Traffic | 33,338 | 2 | 0.005 | 64 | 7038.126 | 0.000 |
| Blog entries | 6.453 | 2 | 0.842 | 64 | 7.668 | 0.001 |
| Social media | 15.755 | 2 | 0.538 | 64 | 29.281 | 0.000 |
| Downloads | 5.233 | 2 | 0.868 | 64 | 6.031 | 0.004 |
| Visit leads | 3.010 | 2 | 0.930 | 64 | 3.236 | 0.046 |
| Purchase leads | 8.775 | 2 | 0.769 | 64 | 11.413 | 0.000 |
| Loyalty leads | 16.587 | 2 | 0.526 | 64 | 31.535 | 0.000 |
| Segment 1: 46; segment 2: 20; segment 3: 1 | | | | | | |

Similarly, it can be seen that in the light of the Anova Test, all the criteria are significantly useful to segment. Therefore, Hypothesis 1 is accepted, and we state that the attraction metric of traffic might be used for describing the segments of the museums' online users. Similarly, Hypotheses 2–5 are confirmed and, hence, we assert that the interest metric of social media links on websites as well as blog entries, download and visits per website might be used for describing the segments of the museums' online users. Likewise, we bear out Hypothesis 6 and, in turn, we claim that the purchase metric of purchasing and booking links per website might be used for describing the segments of the museums' online users. Finally, the positive contrast of Hypothesis 7 allows us to affirm that the metric of loyalty leads per website might be used for describing the segments of the museums' online users.

### 4.3. Cross Tabulation

Finally, we performed a cross-tabulation analysis between the museum segment assignment variables and the island on which the museum is located, and it can be said that there is no significant association between the characteristics of the brick-and-mortar building and the website metrics (see Table 6). However, it stands to reason that as segment three comprises just the Saramago House-Museum, Lanzarote appears associated with the third segment. Nevertheless, we have to reject Hypothesis 8 by pointing out that the island on which the museum is located does not relate to the museum segment metrics of the online users.

**Table 6.** Cross-tabulation analysis between the location (island) and the Canary Islands museum websites segment assignment.

| Islands | | | Segments 1 | 2 | 3 | Total |
|---------|---|---|---|---|---|---|
| El Hierro | | Count | 4 | 1 | 0 | 5 |
| | | % within Island | 80.0% | 20.0% | 0.0% | 100.0% |
| | | % within Cluster | 8.7% | 5.0% | 0.0% | 7.5% |
| | | Adjusted Residual | 0.6 | −0.5 | −0.3 | |
| Fuerteventura | | Count | 5 | 2 | 0 | 7 |
| | | % within Island | 71.4% | 28.6% | 0.0% | 100.0% |
| | | % within Cluster | 10.9% | 10.0% | 0.0% | 10.4% |
| | | Adjusted Residual | 0.2 | −0.1 | −0.3 | |
| Gran Canaria | | Count | 15 | 4 | 0 | 19 |
| | | % within Island | 78.9% | 21.1% | 0.0% | 100.0% |
| | | % within Cluster | 32.6% | 20.0% | 0.0% | 28.4% |
| | | Adjusted Residual | 1.1 | −1.0 | −0.6 | |
| La Gomera | | Count | 1 | 1 | 0 | 2 |
| | | % within Island | 50.0% | 50.0% | 0.0% | 100.0% |
| | | % within Cluster | 2.2% | 5.0% | 0.0% | 3.0% |
| | | Adjusted Residual | −0.6 | 0.6 | −0.2 | |
| Lanzarote | | Count | 4 | 4 | 1 | 9 |
| | | % within Island | 44.4% | 44.4% | 11.1% | 100.0% |
| | | % within Cluster | 8.7% | 20,0% | 100.0% | 13.4% |
| | | Adjusted Residual | −1.7 | 10 | 2.6 | |
| La Palma | | Count | 12 | 4 | 0 | 16 |
| | | % within Island | 75.0% | 25.0% | 0.0% | 100.0% |
| | | % within Cluster | 26.1% | 20.0% | 0.0% | 23.9% |
| | | Adjusted Residual | 0.6 | −0.5 | −0.6 | |
| Tenerife | | Count | 5 | 4 | 0 | 9 |
| | | % within Island | 55.6% | 44.4% | 0.0% | 100.0% |
| | | % within Cluster | 10.9% | 20.0% | 0.0% | 13.4% |
| | | Adjusted Residual | −0.9 | 1.0 | −0.4 | |
| Total | | Count | 46 | 20 | 1 | 67 |
| | | % within Island | 68.7% | 29.9% | 1.5% | 100.0% |
| | | % within Cluster | 100.0% | 100.0% | 100.0% | 100.0% |

| | | Value | Approximate Significance |
|---|---|---|---|
| Nominal by Nominal | Contingency Coefficient | 0.366 | 0.584 |
| | Lambda | 0.014 | 0.314 |
| | Phi and V of Cramer | 0.393 | 0.584 |
| | The Uncertainty coefficient | 0.278 | 0.584 |
| N of Valid Cases | | 67 | |

## 5. Discussions, Implications and Limitations

*5.1. Discussions*

This study aimed to segment the online marketplace of the Canary Islands' museums in accordance with the conversion funnel. Based on previous research [22,29–32], it was expected to segment the online museum marketplace over four phases: awareness, consideration, conversion and loyalty.

The main contribution of this paper is to show the performance of 68 websites of Canary Islands' museums in terms of phases of the conversion funnel. Using metrics of traffic, blog entries, social media, downloads, visit leads, purchase leads and loyalty leads, we placed distributed museums into three categories. Results of the research have confirmed

that chosen metrics as segmentation criteria gave us useful and versatile information that can be used for the segmentation of the online marketplace of the Canary Islands' museums and the development of the website design.

Museums need to rethink their marketing strategy in order to survive changes brought about by COVID-19 [33]. We strongly believe that the current pandemic crisis further increases the importance of segmentation of cultural products based on their website performance. The suggested methodology of the segmentation based on phases of the conversion funnel applies not only to similar cultural organisations but also to other public or private entities.

### 5.2. Implications

The study has managerial relevance as it provides evidence that a conversion funnel is an important tool for the segmentation of the online marketplace based on officially accessed or scrapped data. On this basis, there can be several practical implications to draw as follows:

- The museums' websites lack the development of loyalty tools so that they encourage repeat visitors.
- As far as three different segments have been identified, policymakers might implement specific policies toward each homogenous group so that users' desired responses can be maximized.
- The potential exists to design the mobile application aimed at the evaluation of websites of cultural places. Cultural organisations using the application could check their performance at every stage of the conversion funnel in comparison to the museums that were examined in this research and search for recommendations on how to optimise the website quality.

### 5.3. Limitations

While the current research is actually the first comprehensive study in online museum segmentation with structured data, several limitations must be addressed. First of all, the study was based upon the structured data available on websites or collected with the help of Ubber Suggest software. Although the content analysis was combined with data scraping, "online metrics paint only part of the picture" [87] (p. 21). Therefore, the usage of other qualitative or quantitative techniques is expedient to corroborate the findings and consequently achieve a better data triangulation. Second, the analysis was limited to the museums of the Canary Islands. Although all of them were included in the research, it was still a rather small sample. Third, the statistical analysis did not include the ROPO effect, which might explain the loss of customers throughout the conversion funnel [83,88,89].

In the light of the shortcomings of the current research, we suggest directions for future research. The current research opens up an innovative line of segmentation of the online marketplace, which has the potential to be continued in other regions of Spain or even other countries. The proposed methodology could be easily applied in other public or private contexts as well. Larger-scale research in the segmentation of the online marketplace might benefit the companies managing search engines (Google, Yahoo, Bing, etc.) in drawing the trends of precisely what demands for websites need to be met in order to lead customers from the attraction phase to the loyalty phase. Apart from that, there exists a demand to evaluate the ROPO effect in future research works. This channel-shift pattern can become a real problem for e-commerce companies since its effect suggests that websites are used as a mere showcase or as a simple basic channel to acquire information but not as a purchase channel [83,88,89]. Some authors even dare to classify this phenomenon as a true enemy for the development of online stores [90,91]. Consideration of the ROPO effect could enable marketers of museums to better understand various leakages between the different phases of the conversion funnel and identify customer groups that are most responsive to marketing activities.

## 6. Conclusions

This paper concerns an important problem of segmenting the online marketplace of the Canary Islands' museums by using different conversion funnel metrics. The research allowed us to identify the segment of museums that demonstrate insufficient activity in proceeding from the attraction to the loyalty phase. The traffic of museums belonging to the second segment shows room for improvement in strengthening market position. Based on the results of the study, it should also be stated that in terms of traffic, blog entries, visit leads and loyalty leads, the performance of the Saramago museum in Lanzarote (third segment) was found to be outstanding. We hope that managers of museums in the Canary Islands may glean the lessons from the Saramago museum in advancing the development agenda.

The presented way to apply the conversion funnel model has shown that the model can be a valid foundation point for the attraction of new customers to museums. Moreover, it is an efficient tool to increase the share of loyal customers of cultural organizations currently challenged by the COVID-19 pandemic.

The authors verified the applicability of the conversion funnel model for the segmentation of the online marketplace based on primary and secondary data officially accessed on websites. The possibility to segment the online marketplace without direct contact with representatives of museums or their customers is extremely important under the conditions of the pandemic.

This paper shows several limitations. First, although three segments are identified, they are quite different in size. What is more, one of the segments only comprises one unit. In fact, we regret that post hoc tests were not performed because the third segment has just one case and museum, and therefore no more ground can be found to validate statistically this triple segment structure. Needless to say, it is not ideal from the cluster assumptions. Nevertheless, this market structure was chosen because, from a managerial point of view, it seems interesting to implement specific marketing policies toward the most important museum of The Canary Islands. Regardless of a strict statistics demand, it can be said that it is not the sizes of the segments that matter most but rather the quality of the strategic and managerial approaches. Be that as it may, the standardisation of the used variables and the randomised survey strengthens the cluster statistical assumptions. Second, while we used software for measuring the website traffic, we checked the rest of the metrics manually. All the websites were analysed with the same software at the same time. In both cases, that is, using the software and operating manually, the variables are quantitative and objective. Therefore, our approach was merely quantitative because we did not assess the quality of the websites nor any other subjective approach. For this reason, a more qualitative approach is needed, for example, assessing the user experience by considering emotional dimensions such as satisfaction, friendliness and empathy. Finally, it is worth acknowledging that as we are not webmasters of the museums' websites, we could not extract exact metrics. Therefore, although we performed the web-analytic task under similar circumstances, that is, with the same software for all the museums at the same time and under one supervisor, the reliability and validity of the extracted metrics must not be 100%.

**Author Contributions:** Conceptualization, G.D.M.; methodology, G.D.M.; software, G.D.M., M.E.O. and N.V.-V.; validation, G.D.M.; formal analysis, G.D.M.; investigation, G.D.M., M.E.O. and N.V.-V.; resources, G.D.M.; data curation, G.D.M.; writing—original draft preparation, G.D.M., M.E.O. and N.V.-V.; writing—review and editing, G.D.M., M.E.O. and N.V.-V.; visualization, G.D.M.; supervision, G.D.M.; project administration, G.D.M.; funding acquisition, G.D.M. All authors have read and agreed to the published version of the manuscript.

**Funding:** Although this research received no external funding, it is developed under the Research Chair of Social Innovation and Digital Transformation for the Culture and Arts in The Canary Islands.

**Data Availability Statement:** Please contact the corresponding author if needed.

**Conflicts of Interest:** The authors declare no conflict of interest.

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
