# Peer review of "Online Museums Segmentation with Structured Data: The Case of the Canary Island’s Online Marketplace"

_jtaer, doi:10.3390/jtaer16070151_

Round 1
Reviewer 1 Report
Online Museums Segmentation with Structured Data: the Case of the Canary Island’s Online Marketplace
Before reviewing the paper further and preparing it for possible publication, I have the following comments to the authors to improve their paper and resubmit:
This paper’s primary objective is to gain an understanding of the online users of The Canary Islands’ museum websites.
- The Literature Review seems outdated, given that the topic is innovative and up to date.
- The purpose is not clear! “… to gain an understanding of the online users” is not specific enough. What kind of understanding? Understanding of what?
- The authors should improve the Introduction Section greatly to state the motivation of their study clearly, tell readers why their study is important and state their contribution to the literature clearly.
- The literature review should lead the reader to support the methodology section of the study. For example, what studies in the past have used the same methodology? What was their sample size? How did they collect the data? What are the choices of instrumentation? Why did the authors decide to use the current instrumentation and research methodology?
- Have there been any studies related to “the conversion funnel, also referred to as a marketing funnel” since a decade ago?
- What is the use of Table 1 for non-Spanish speaking readers?
- There are inconsistencies with numerical representation of “,” and “.” (See table 4 Traffic, and then Social Media percentages. I am confused when “.” is used in the American way and when it is used in a European way. Table 6 goes back to sue of “,’ as a decimal separator for percentages.
- Here are some of the conclusions:
- There are social media platforms and media downloads whose potential interaction is not exploited. (L330)
- There is room to improve the sales orientation of the Canary Island museum (L331) marketplace.
- There are resources, such as online traffic, that have not been tapped to enhance (L327) engagement and participation through blogs.
- These conclusions are so general and so vague that you do not need a study for them. Furthermore, we can state these conclusions about virtually any business. It is like saying: “there is always room for improvement.”
- What are the implications of this study?
- What are the hypotheses?
- What is the motivation behind this study?
Author Response
Reviewer 1
Dear Reviewer 1, We would like to thank you for your review and time. Below you can find a nuanced response to your comments.
|
· Reviewer’s comments |
Authors’ comments |
||
|
· The Literature Review seems outdated, given that the topic is innovative and up to date. |
The literature review has been supplemented with additional texts and up-to-date research works in this field. We hope we meet your expectations. |
||
|
The purpose is not clear! “… to gain an understanding of the online users” is not specific enough. What kind of understanding? Understanding of what? |
The purpose has been revised so that we hope it is more clear and consistent. The purpose is “to segment the online marketplace of the Canary Islands’ museums by using different conversion funnel metrics”. |
||
|
The authors should improve the Introduction Section greatly to state the motivation of their study clearly, tell readers why their study is important and state their contribution to the literature clearly. |
The introduction has been improved. We hope that the current version of the introduction reveals the relevance of the research, motivation of the study, contribution to the literature and practice. |
||
|
The literature review should lead the reader to support the methodology section of the study. For example, what studies in the past have used the same methodology? What was their sample size? How did they collect the data? What are the choices of instrumentation? Why did the authors decide to use the current instrumentation and research methodology? |
To the best of our knowledge, no studies in the past have used the same methodology for the segmentation of organisations according to their websites. Despite that, we have improved the literature review with additional reasoning for the choice of metrics used for segmentation. Hopefully, it leads the reader to support the methodology section of the study. |
||
|
Have there been any studies related to “the conversion funnel, also referred to as a marketing funnel” since a decade ago? |
There have been some studies; they are mentioned in the improved version of the introduction. |
||
|
What is the use of Table 1 for non-Spanish speaking readers? |
Thank you for this suggestion. Although Tables 1 & 2 comprise the original museums’ name, we have translated it into English so that the reader can understand the original name. |
||
|
There are inconsistencies with numerical representation of “,” and “.” (See table 4 Traffic, and then Social Media percentages. I am confused when “.” is used in the American way and when it is used in a European way. Table 6 goes back to sue of “,’ as a decimal separator for percentages. |
We have revised the inconsistencies so that the percentages presented in tables 4 and 6 are more consistent.
|
||
|
· Here are some of the conclusions: · There are social media platforms and media downloads whose potential interaction is not exploited. (L330) · There is room to improve the sales orientation of the Canary Island museum (L331) marketplace. · There are resources, such as online traffic, that have not been tapped to enhance (L327) engagement and participation through blogs. These conclusions are so general and so vague that you do not need a study for them. Furthermore, we can state these conclusions about virtually any business. It is like saying: “there is always room for improvement.” |
Thank you so much for the insights. We have waived these statements. |
||
|
· What are the implications of this study? |
The implications of the study are provided in the Discussion whose insight has been improved significantly. |
||
· What are the hypotheses? |
We have formulated and tested some hypotheses in the review of the literature and the analysis of the results sections, respectively. These hypotheses are as follows:
H1: The attraction metric of traffic might be used for describing the segments of the museums' online users.
H2: The interest metric of social media links on websites might be used for describing the segments of the museums' online users. H3: The interest metric of blog entries per website might be used for describing the segments of the museums' online users. H4: The interest metric of download per website might be used for describing the segments of the museums' online users. H5: The interest metric of visits leads per website might be used for describing the segments of the museums' online users.
H6: The purchase metric of purchasing and booking links per website might be used for describing the segments of the museums' online users
H7: The metric of loyalty leads per website might be used for describing the segments of the museums' online users.
H8: the island on which the museum is located relates to the museum segment metrics of the online users.
|
|
||
|
· What is the motivation behind this study? |
The motivation behind this study is revealed in the improved version of the Introduction. |
||
Let us reiterate our gratitude to you for your time and comments.

Reviewer 2 Report
The idea of the study is very interesting and up-to-date, however, the empirical part of the article, which uses statistical methods, requires more exploration by the authors of the methodological aspects of statistical tools used in the study (e.g. criteria for selecting the value k parameter in the k-means method or ANOVA assumptions). To use statistical methods, you need to know not only how to calculate, but also whether a given method can be applied (e.g. check the method assumptions).
Detailed comments:
- Page 5, line 239. How many objects are there in the population? Here it is written 155, while in tab. 3 (page 8) 160 museums are listed.
- Page 6, line 263. How many objects does the sample count? Here it is written 68, whereas in tab. 4 (page 9, line "Download") or in the statistical analyzes included 67.
- Page 6, line 266. "And" instead of "y".
- Page 8, line 274. “Descriptive statistics” instead of “Descriptive statistical test”.
- Page 8, line 277. "Percent" instead of "per cent".
- Page 9, Fig. 1. The elements of the "Interest" section should be arranged in the same order as in tab. 4. (page 9).
- Page 9, line 281. I think that the correct sentence should be: "Regarding interest it seems clear that ..." (instead of: "Regarding the interest, purchasing and loyalty rates, it seems clear that ..."), because later in this sentence the conclusions concern only the variables included at this (interest) stage of the conversion funnel (social media links, downloads, visit forms).
- Page 9, line 286. Description of 4: "Frequency and descriptive statistics of the Canary Islands museum websites" (instead of "Frequency and descriptive data on…”).
- Page 10, line 298 and 306. Tab. 5 and tab. 6 have the same title. Moreover, this title does not de facto explain what the results are (what method it comes from).
- Page 10, tab 6. Discard the symbol “V3”; leave only "Island".
Methodological notes on Section 4.2.
- I have doubts about the use of ANOVA results as a criterion for selecting the value of the k parameter (number of groups). The ANOVA method has a lot of restrictive assumptions (e.g. the normality distribution of variables in groups), which, if not met, may lead to wrong conclusions when using this method. The ANOVA table can only be used as an aid when trying to determine the discriminant power of variables, while the selection of the number of groups is based on special criteria. In this case it may be internal measures for cluster validation, e.g. Calinski-Harabasz index, Davies-Bouldin index, Dunn index and many others .
- As a result of the conducted analysis, a single-element group was extracted (Saramago Museum). Did the authors of the study, prior to analyzing the data using the k-means method, trace the values of the variables in order to look for atypical observations (outliers or extreme observations which are usually excluded from this type of analysis)? Very high values of final cluster centers for traffic, social media and loyalty leads variables suggest caution when including this object in the analysis.
- Conclusions from the results obtained by the k-means method:
- Page 10, line 294-5. Why was loyalty leads omitted? The value of the cluster center coordinate for this variable is higher than, for example, for downloads.
Methodological notes on Section 4.3.
- What specific coefficient was calculated, because in the case of nominal variables there are many of them (eg.: C Pearson, V Cramer, Czuprow).
- Calculating the contingency coefficient, when so many cells in the multi-way table are equal 0, is incorrect. The measures of correlation for nominal features are based on the chi-square statistic, which sets requirements as to the minimum number of cells in the contingency table (minimum 5).
Author Response
Reviewer 2
Dear Reviewer 2, We would like to thank you for your review and time. Below you can find a nuanced response to your comments.
· Reviewer’s comments |
Authors’ comments |
The idea of the study is very interesting and up-to-date, however, the empirical part of the article, which uses statistical methods, requires more exploration by the authors of the methodological aspects of statistical tools used in the study (e.g. criteria for selecting the value k parameter in the k-means method or ANOVA assumptions). To use statistical methods, you need to know not only how to calculate, but also whether a given method can be applied (e.g. check the method assumptions). |
Thank you for your point. In effect, the survey procedure is probabilistic at random and, hence, one of the Cluster and ANOVA requirements is met. In addition, we can state that, before operating with the data, we standardised the values of the variables. Therefore, K means, as well as Anova, work with a normal distribution. Nevertheless, we cannot state that the obtained clusters are of a similar size because not only are the two main segments different in size but the third segment comprises 1 unit. We have no choice but to acknowledge it as a limitation in the conclusion section, even though we think it is worth keeping this segmented structure for management sake.
|
Page 5, line 239. How many objects are there in the population? Here it is written 155, while in tab. 3 (page 8) 160 museums are listed. |
Thank you so much. We have amended the error. The final population is 155 and it is revised in the survey spreadsheet (table 3).
|
Page 6, line 263. How many objects does the sample count? Here it is written 68, whereas in tab. 4 (page 9, line "Download") or in the statistical analyses included 67. |
Thank you so much. This inconsistency is due to a missing value. Although the final sample comprises 68 cases, there is a museum without data related to its download link. As a consequence, for the download link the sample is 67. Unfortunately, if we check this museum website now, we cannot know whether at the time of the survey the website was the same. For this reason, now we think that adding the value would create a new inconsistency in the database stemming from different time frameworks. |
Page 6, line 266. "And" instead of "y". |
This grammatical error has been corrected. |
Page 8, line 274. “Descriptive statistics” instead of “Descriptive statistical test”. |
We appreciate the suggestion and we have modified it. |
Page 8, line 277. "Percent" instead of "per cent". |
This grammatical error has been corrected. |
Page 9, Fig. 1. The elements of the "Interest" section should be arranged in the same order as in tab. 4. (page 9). |
Thanks for the observation. The elements have been arranged in the same order (fig 1). |
Page 9, line 281. I think that the correct sentence should be: "Regarding interest, it seems clear that ..." (instead of: "Regarding the interest, purchasing and loyalty rates, it seems clear that ..."), because later in this sentence the conclusions concern only the variables included at this (interest) stage of the conversion funnel (social media links, downloads, visit forms). |
Thank you so much for your positive proposition. We agree and the correct sentence has been applied. |
Page 9, line 286. Description of 4: "Frequency and descriptive statistics of the Canary Islands museum websites" (instead of "Frequency and descriptive data on…”). |
Thanks for the suggestion. We have modified it. |
Page 10, lines 298 and 306. Tab. 5 and tab. 6 have the same title. Moreover, this title does not de facto explain what the results are (what method it comes from). |
Thank you very much for paying attention to this error. We have modified and adapted the titles of table 5 and table 6 so that the titles refer to the content of each one. |
Page 10, tab 6. Discard the symbol “V3”; leave only "Island". |
The V3 symbol has been removed. |
I have doubts about the use of ANOVA results as a criterion for selecting the value of the k parameter (number of groups). The ANOVA method has a lot of restrictive assumptions (e.g. the normality distribution of variables in groups), which, if not met, may lead to wrong conclusions when using this method. The ANOVA table can only be used as an aid when trying to determine the discriminant power of variables, while the selection of the number of groups is based on special criteria. In this case, it may be internal measures for cluster validation, e.g. Calinski-Harabasz index, Davies-Bouldin index, Dunn index and many others. |
We are afraid that we have not been able to perform post hoc tests because one of the segments has only one case. For this reason, we have recognised it as a limitation in the conclusion section. In addition, following the reviewer suggestions, we have put forward some hypotheses in whose empirical contrast the Anova test plays a role to determine the discriminant power of the variables. Nevertheless, to strengthen the decision of identifying three different segments, we have taken several measures. First, we carried out a discriminant test and we can state that this triple segment structure comes up with 100% of the cases correctly classified. Second, we checked the iteration history and it can be said that, after six iterations, the convergence is achieved as well as the minimal distance between initial centres is 8.9. Third, we would like to highlight the importance of managerial criteria since we think that three segments are the best solution by considering both market and practical implications. Finally, we performed similarity (Pearson correlation) and dissimilarity (Euclidian distance) tests to check whether the distances between every couple of cases are consistent with the segment assignation. |
As a result of the conducted analysis, a single-element group was extracted (Saramago Museum). Did the authors of the study, prior to analysing the data using the k-means method, trace the values of the variables in order to look for atypical observations (outliers or extreme observations which are usually excluded from this type of analysis)? Very high values of final cluster centres for traffic, social media and loyalty leads variables suggest caution when including this object in the analysis. |
Thank you for your insightful point. We agree that that the Saramago Museum represents what can be considered atypical and outlier in the Canary Islands. It is the most international and remarkable museum on the internet for this destination. As it is known, Saramago was a Nobel writer who chose Lanzarote Island as home and his reputation is global. This is the reason for obtaining such a different positioning on the internet although possibly this museum is not the most visited neither popular offline. We did not eliminate the museum from the database because of Its global reputation and importance worth a specific treatment. Moreover, managing this specific museum with specific digital marketing strategies might be cost worthy and profitable. No doubt, The Canary islands government shows enough capacity to develop and implement specific policies for this museum. To put it simply, as far as identifying the market segment structure is concerned it is not the museums units number that matter most but rather the strategical importance and significant positioning of the museums included in the segment. |
Conclusions from the results obtained by the k-means method: Page 10, line 294-5. Why was loyalty leads omitted? The value of the cluster centre coordinate for this variable is higher than, for example, for downloads. |
Please, forgive us if we misunderstand your comment concerning table 4, but there is one Download that is a missing value. This is the reason for being 67 cases related to the download leads variables instead of 68 cases as traffic, blog entries, social media, visit leads, purchase leads and loyalty variables are. Hence, it was not omitted but rather it is a missing value.
Concerning table 5, we can state that the reason for a higher value at the loyalty leads variable than at the download variable is because these values are standardised. In other words, we standardised the values to make them more easily understandable and interpretable. Therefore, data in Tables 4 and 5 do not show the original obtained values but rather standardised values. |
What specific coefficient was calculated, because in the case of nominal variables there are many of them (eg.: C Pearson, V Cramer, Czuprow). Calculating the contingency coefficient, when so many cells in the multi-way table are equal 0, is incorrect. The measures of correlation for nominal features are based on the chi-square statistic, which sets requirements as to the minimum number of cells in the contingency table (minimum 5). |
In addition to the contingency coefficient, we have included other metrics such as Lambda, Phi and V of Cramer, and the uncertainty coefficient. |
Let us reiterate our gratitude to you for your time and comments.

Reviewer 3 Report
Dear Authors,
Thank you for your manuscript and please find my remarks and comments.
The presented paper deals with an important topic. First: online behaviour of consumers still need to be investigated, and especially in terms of cultural goods the number of research is limited. Second: a website might be a powerful tool to promote culture and the value of regional brands.
The strengths of the paper are as follows:
- important topic
- attempt to use a theoretical framework to conduct a segmentation
However, in my opinion, the whole paper needs to be carefully reconsidered and restructure. I will present all my doubts/recommendations in the following sections:
Section 1: the aim of the study
The aim indicated in the introduction is “This paper’s primary objective consists of segmenting the online marketplace of the Canary Islands’ museums by using different conversion funnel metrics. These” – what could be clear, however in the Abstract: we can find “This paper’s primary objective is to gain an understanding of the online users of The Canary Islands’ museum websites.”
As you can see those aims are very different – I think the relevant aim is mentioned in the introduction, while declaration in the abstract is rather a mistake taking into consideration the whole content of the proposed manuscript.
Taking into account the content of the manuscript the aim is rather websites assessment than segmentation.
Section 2: literature review
The Authors focused on the “conversion channel”. Taking into consideration the topic of the paper and object of the research I missed, e.g.:
- references to any other research related to the websites assessment / segmentation
- consumption and promotion of cultural goods
Additionally, it is not clear why the ROPO effect is elaborated as I could not find any reference to that topic in the results and discussion part.
Therefore in my opinion the literature part needs to be restructured and supplemented.
Section 3: a methodological approach
The conversion model could be a framework for segmentation, however, it is very difficult to separate the assessment of the website itself from the assessment and perception of the museum. For instance:
In terms of attraction – you proposed to use “traffic and the number of users visiting the websites monthly” => this indicator is rather related to the museum (how well known it is) but not to the website itself.
For the other components of the conversion channel, you used metrics like: “existence of blog, social media and download resources” etc. It means you considered only “existence” and did not evaluate the quality of that, e.g. how user-friendly it is?
Section 4: results
It is rather questionable to put just one object in the cluster what is actually a case for “Saramago”.
Section 5: more specific remarks
Table 1. El Hierro’s, Fuerteventura, La Gomera, Lanzarote y La Palma museums. => Is table 1 really necessary in the main body of the paper?
Table 6. Some noted studies on consumer attitudes toward CRM. => how the title refers to the content of the table
Author Response
Reviewer 3
Dear Reviewer 3, We would like to thank you for your review and time. Below you can find a nuanced response to your comments.
· Reviewer’s comments |
Authors’ comments |
Thank you for your manuscript and please find my remarks and comments. The strengths of the paper are as follows: However, in my opinion, the whole paper needs to be carefully reconsidered and restructure. I will present all my doubts/recommendations in the following sections: |
Thank you so much for your positive and critical comments. It has encouraged us to move forward and we have done our best to improve our paper by following your indications. |
Section 1: the aim of the study |
Thank you so much for paying attention to this incongruity. We have specified the aim in the abstract. |
Section 2: literature review |
We have added references to up-to-date research works in the field of websites assessment and segmentation based on the conversion funnel. |
- consumption and promotion of cultural goods. |
We have added some content related to cultural goods (where it was possible to find relevant research works). |
Additionally, it is not clear why the ROPO effect is elaborated as I could not find any reference to that topic in the results and discussion part. |
We have added some consideration about the ROPO effect in the Discussion part as well as we have given more emphasis on the interplay between online and offline contexts in the review of the literature. Now the ROPO effect is a future line of research. |
Therefore in my opinion the literature part needs to be restructured and supplemented. |
We have restructured the literature review as it consistently leads to the research methodology. In addition, we have supplemented the literature review with up-to-date research works to support the new hypotheses. |
Section 3: a methodological approach |
While we used software for measuring the website traffic, we checked the rest of the metrics manually. All the websites were analysed with the same software at the same time. In both cases, that is, using the software and operating manually, the variables are quantitative and objective. Therefore, we recognise humbly that our approach was merely quantitative because we did not assess the quality of the websites nor any other subjective approach. We have appended it as a limitation to the conclusion section and in line with it, we have proposed future lines of research.
|
Section 4: results |
Thank you for your insightful point. We agree that that the Saramago Museum represents what can be considered atypical and outlier in the Canary Islands. This is because this is the most international and remarkable museum on the internet for this destination. As it is known, Saramago was a Nobel writer who chose Lanzarote Island as home and his reputation is global. This is the reason for obtaining such a different positioning on the internet although possibly this museum is not the most visited neither popular offline. We did not eliminate the museum from the database because of Its global reputation and importance are worth a specific treatment. It is the most international online museum on Canary Island. Moreover, dealing with this specific museum with specific digital marketing strategies might be cost worthy and profitable. No doubt, The Canary islands government shows enough capacity to develop and implement specific policies for this museum whose support is widely accepted. To put it simply, as far as identifying the market segment structure is concerned, it is not the museums´ units numbers that matter most but rather the strategical importance and significant positioning of the museums included in the segment. |
Section 5: more specific remarks Table 1. El Hierro’s, Fuerteventura, La Gomera, Lanzarote y La Palma museums. => Is table 1 really necessary in the main body of the paper? |
We have translated table 1 into English and revised table 6 title so that it is more specific to its content. Thank you very much for your comment. In our view, it can be interesting to know the museums' names since the number of museums is not as high as to make it impossible. |
Let us reiterate our gratitude to you for your time and comments.

Round 2
Reviewer 1 Report
Thank you for revising the paper. The paper has been improved. Thank you for your clear and precise responses. However, there are still certain points:
The implications of the study are provided in the Discussion whose insight has been improved significantly.
I suggest creating a separate heading to indicate your implications. 5. Discussions, implications and limitations. Then 5.1 Discussions; 5.2 Implications; 5.3 Limitations
Table 4. Frequency and descriptive statistics of the Canary Islands museums websites
STD of traffic missing separators! "1742765,305"
Furthermore, it is worth noting that the minimal distance obtained (L368) between initial centroids is 8.9.
What is 8.9? is it 8,9? Where in table 5 can I find this number?
The argument to base H2 to H5 is very brief and ONLY based on TWO references?
Author Response
Dear Reviewer 1, let us express our gratitude for your review and time. Please, below, you can find the response to your comments.
Reviewer’s comments |
Authors’ comments |
Thank you for revising the paper. The paper has been improved. Thank you for your clear and precise responses. However, there are still certain points: |
Thank you so much for providing us with a helpful review and your comments. We hope we make good use of your feedback. |
The implications of the study are provided in the Discussion whose insight has been improved significantly. I suggest creating a separate heading to indicate your implications. 5. Discussions, implications and limitations. Then 5.1 Discussions; 5.2 Implications; 5.3 Limitations |
Thank you. We have revised the headings as you have suggested. |
Table 4. Frequency and descriptive statistics of the Canary Islands museums websites. STD of traffic missing separators! "1742765,305" |
Thank you for paying attention to this. We have realised that there were some inconsistencies as regards the way we wrote the numbers. We have revised the whole manuscript and corrected this kind of mistake. |
Furthermore, it is worth noting that the minimal distance obtained (L368) between initial centroids is 8.9. What is 8.9? is it 8,9? Where in table 5 can I find this number? |
Thank you. These inconsistencies and mistakes have also been amended. |
The argument to base H2 to H5 is very brief and ONLY based on TWO references? |
Thank you. We have strengthened the literature to support these hypotheses. Right now, there are new references to back them up. |
Let us reiterate our gratitude for your help in reviewing our paper

Reviewer 3 Report
Dear Authors,
Thank you for your updated manuscript, and please find my remarks and comments.
As this is the second review of the paper (after update done by the Authors) I put some general comments and refer to the parts changed by the Authors. I find the current version of the paper significantly improved.
Aim o the paper
The aim of the study is much more clear and aim declaration in the abstract and introduction fits each other. Additionally, you have put hypothesis what I perceived as reasonable.
Methods and data analysis
This section is just slightly changed.
You answered to my comment as follows: “While we used software for measuring the website traffic, we checked the rest of the metrics manually. All the websites were analysed with the same software at the same time. In both cases, that is, using the software and operating manually, the variables are quantitative and objective. Therefore, we recognise humbly that our approach was merely quantitative because we did not assess the quality of the websites nor any other subjective approach. We have appended it as a limitation to the conclusion section and in line with it, we have proposed future lines of research.”
I agree with your comment that you focus on the quantitative approach. However in the first part of my remark I mentioned “it is very difficult to separate the assessment of the website itself from the assessment and perception of the museum. For instance: In terms of attraction – you proposed to use “traffic and the number of users visiting the websites monthly” => this indicator is rather related to the museum (how well known it is) but not to the website itself.” I understand that is not possible to change measures at this stage, however I miss your reply on that point.
Segmentation
I accept your explanation concerning Saramago Museum.
Good luck with your revision
Reviewer
Author Response
Dear Reviewer 3, thank you very much indeed for your new review and time. The response to your comments is below:
Reviewer’s comments |
Authors’ comments |
Dear Authors, thank you for your updated manuscript, and please find my remarks and comments. As this is the second review of the paper (after update done by the Authors) I put some general comments and refer to the parts changed by the Authors. I find the current version of the paper significantly improved. Aim of the paper. The aim of the study is much more clear and aim declaration in the abstract and introduction fits each other. Additionally, you have put hypothesis what I perceived as reasonable. |
Thank you so much for your positive remarks. We have done our best to improve our paper by following your suggestions. |
Methods and data analysis. This section is just slightly changed. You answered to my comment as follows: “While we used software for measuring the website traffic, we checked the rest of the metrics manually. All the websites were analysed with the same software at the same time. In both cases, that is, using the software and operating manually, the variables are quantitative and objective. Therefore, we recognise humbly that our approach was merely quantitative because we did not assess the quality of the websites nor any other subjective approach. We have appended it as a limitation to the conclusion section and in line with it, we have proposed future lines of research.” I agree with your comment that you focus on the quantitative approach. However in the first part of my remark I mentioned “it is very difficult to separate the assessment of the website itself from the assessment and perception of the museum. For instance: In terms of attraction – you proposed to use “traffic and the number of users visiting the websites monthly” => this indicator is rather related to the museum (how well known it is) but not to the website itself.” I understand that is not possible to change measures at this stage, however I miss your reply on that point. |
Thank you for your reflection. We agree with you that it is hard to distinguish the assessment of the website itself and the museum itself. We considered this idea a bit when we created hypothesis 8 since we state that online and offline realities are interplaying. However, we only mean online traffic and online users nothing to do with the offline museum when we say attraction. But you are right when you suggest that we should carefully consider the indicator because the software is not 100% reliable because it is not working from inside the website but outside. In other words, we are not webmasters of the museums’ website, neither did we set up google analytics. However, we monitored all the museums with the same software during the same period and under similar circumstances. The measurement procedure worked under the influence of the same measuring instrument and, hence, we limited the variation of external and internal variables. To take advantage of your question, we have appended it as a limitation to the paper as follows: “Finally, it is worth acknowledging that as we are not webmasters of the museums’ websites, we could not extract exact metrics. Therefore, although we performed the web-analytic task under similar circumstances, that is, with the same software for all the museums during the same time and under one supervisor, the reliability and validity of the extracted metrics must not be 100%.” |
Segmentation. I accept your explanation concerning Saramago Museum. Good luck with your revision |
Thank you so much for your understanding. |
Let us reiterate our gratitude for your help in improving our paper

Round 3
Reviewer 1 Report
Thank you for your responses. I am having serious doubts about "The argument to base H2 to H5 is very brief and ONLY based on TWO references?" You have added a couple of references that are not from credible sources. There seems to me you do not have enough support for your hypotheses. I will give you another chance to show support from the literature review for EACH hypothesis separately.
Author Response
Dear Reviewer 1, we are grateful for giving us these pointers. Please, find below the response to your comments.
Reviewer’s comments |
Authors’ comments |
Thank you for your responses. |
Welcome and thank you very much indeed for your review and time. |
I am having serious doubts about "The argument to base H2 to H5 is very brief and ONLY based on TWO references?" You have added a couple of references that are not from credible sources. There seems to me you do not have enough support for your hypotheses. |
We have gone back over these arguments to supplement hypotheses 2 & 5 with more references. In addition to adding more references to hypotheses 2 & 5, we have also revised how we had supported hypotheses 3, 4, and 6 so that more citations and rationale can be found. |
I will give you another chance to show support from the literature review for EACH hypothesis separately.
|
We have separated all the hypotheses to supporting each of them specifically and separately. |
Kind regards